# A mathematical model that predicts human biological age from physiological traits identifies environmental and genetic factors that influence aging

**Sergiy Libert\*, Alex Chekholko, Cynthia Kenyon\***

Calico Life Sciences, South San Francisco, United States

## eLife Assessment

This **important** study developed a mathematical model to predict biological age by leveraging physiological traits across multiple organ systems. The results presented are **convincing**, utilizing comprehensive data-driven approaches. However, additional external validation could further strengthen its generalizability. The model provides a way to identify environmental and genetic factors impacting aging and lifespan, revealing new factors potentially affecting aging. It also shows promise for evaluating therapeutics aimed at prolonging a healthy lifespan.

**\*For correspondence:**
libert@calicolabs.com (SL);
cynthia@calicolabs.com (CK)

## Abstract

Why people age at different rates is a fundamental, unsolved problem in biology. We created a model that predicts an individual's age from physiological traits that change with age in the large UK Biobank dataset, such as blood pressure, lung function, strength, and stimulus-reaction time. The model predicted a person's age with best accuracy when it heavily weighted traits that together query multiple organ systems, arguing that most or all physiological systems (lung, heart, brain, etc.) contribute to the global phenotype of chronological age. Differences between calculated 'biological' age and chronological age (ΔAge) appear to reflect an individual's relative youthfulness, as people predicted to be young for their age had a lower subsequent mortality rate and a higher parental age at death, even though no mortality data were used to calculate ΔAge. Remarkably, the effect of each year of physiological ΔAge on Gompertz mortality risk was equivalent to that of one chronological year. A genome-wide association study (GWAS) of ΔAge and analysis of environmental factors associated with ΔAge identified known as well as new factors that may influence human aging, including genes involved in synapse biology and a tendency to play computer games. We identify a small number of readily measured physiological traits that together assess a person's biological age and may be used clinically to evaluate therapeutics designed to slow aging and extend healthy life.

## Introduction

The process of aging is universally similar yet deeply unique to each person. By observing a person for a moment, one can deduce their age with high accuracy, even though no two people age the same way. Some individuals might lose hair with age or develop chronic diseases, whereas others might not. Investigating both the universal aspects of aging and the basis of individual differences, and developing means of measuring physiological age and health, will provide opportunities to improve human lives.

The rate of aging, i.e., the rate at which organisms lose physiological fitness and accumulate morbidity, has both genetic and environmental determinants. Humans age more slowly than, for example, dogs, so genes play a key role, but environmental factors like smoking and exercise influence aging as well. In this study, we have used publicly available data of human health parameters to systematically identify genetic and environmental variables that influence human aging.

To generate an inclusive, holistic model of human aging, we queried a large, well-annotated human database (the United Kingdom BioBank [UKBB]) comprising over 3000 phenotypes that together span the functions of multiple organs and physiological systems. The UKBB's medical, environmental, and genetic data on ~500,000 British volunteers is a unique resource to investigate the biology of aging. While participants of UKBB are not a random cross section of society (*Abdellaoui et al., 2019*; *Haworth et al., 2019*), this rich database nonetheless likely provides generalizable insights into human aging and disease (*Hanlon et al., 2022*).

A number of published studies describe and employ methods to identify genes that might influence human aging. The majority of those studies (*Ruby et al., 2018*; *Pilling et al., 2017*; *Wright et al., 2019*) focus on lifespan (*Joshi et al., 2017*; *Timmers et al., 2019*; *Timmers et al., 2022*), e.g., age at death or parents' age at death, or analyze cohorts of people with exceptional lifespan (*Bae et al., 2022*; *Shen et al., 2020*), or the presence or absence of one or few age-associated diseases (*Timmers et al., 2022*; *Timmers et al., 2020*; *Zenin et al., 2019*). Additionally, researchers have used molecular traits, such as blood proteins (*Coenen et al., 2023*) or blood DNA methylation patterns, to build and analyze biological age prediction algorithms (clocks) to identify genes that influence aspects of human aging (*Gibson et al., 2019*; *Lu et al., 2018*; *McCartney et al., 2021*). Biological age clocks derived from one or few physiological measures have also been constructed, such as a biological clock built using 3D facial scans (*Xia et al., 2020*). Likewise, a biological clock built using the gut microbiome (*Wilmanski et al., 2021*) was used to identify individuals who might be aging slower or faster than average and suggest drugs that might influence gut health. Recently, aging of separate organs has been investigated and linked to age-associated diseases and mortality (*Tian et al., 2023*), and biological age has been estimated using AI methods (*Qiu et al., 2022*).

In our approach, we set out to measure human aging directly and holistically, making sure that all systems relevant to health are represented. To do so, we sampled and analyzed traits reporting on multiple organ systems and physiological domains. We quantified the markers of aging that reflect overall physiological health, such as strength, stimulus-reaction time, and blood pressure. This multi-systemic approach does not rely on the presence or absence of recognized diseases or a small number of binary events, such as death or stroke, and therefore reflects human aging more directly. Likewise, instead of concentrating on diseases, we aimed to evaluate a multitude of physiological parameters that change in 'healthy' people, to allow us to identify factors missed by previous studies.

We developed a series of mathematical models that consider 121 age-related traits and predict a biological age for each individual. We show that the model that best predicts age incorporates data reflecting the activity of most, if not all, the organs and physiological systems. By comparing predicted biological age to actual age, we identified individuals who may be aging slower or faster than average. Using this model, we identified new environmental factors and genetic loci that may influence biological age. By building models lacking clusters of phenotypically correlated (typically organ-specific) traits, we further categorized these genetic loci and environmental factors as those likely to influence aging globally vs those that likely impact a single organ system. Likewise, by analyzing a smaller, healthier sub-cohort of UKBB participants, we identified factors likely to influence apparent age by conferring an age-related disease. Notably, our findings highlighted neural function as an important determinant of overall biological age. Finally, after analyzing the performance of different physiological clocks, we identified 12 key physiological traits that together could measure biological age in longitudinal clinical trials for interventions that increase human health span.

## Results

### Physiological traits that change with age

To identify age-dependent traits, we conducted linear regression analysis on every UKBB parameter relative to the age of the participants (see *Supplementary file 1* and *Supplementary file 2*) and recorded the list of those with a non-zero slope and adjusted statistical significance better than $10^{-3}$

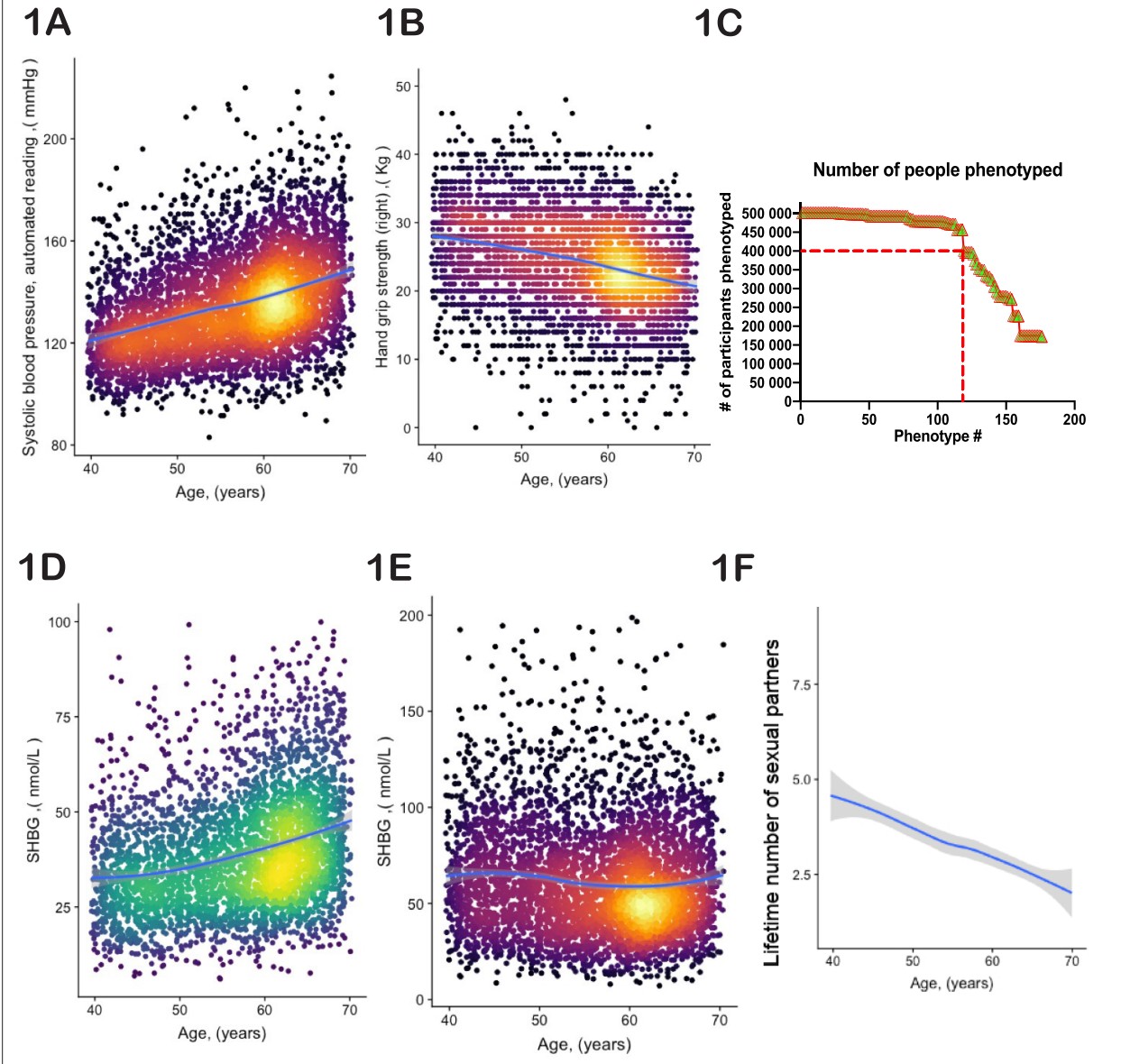

**Figure 1.** Selection of physiological phenotypes for bioloigcal age modelling. (**A**) Systolic blood pressure (United Kingdom BioBank [UKBB] field ID# 4080) and (**B**) hand-grip strength (UKBB field ID# 47) of a random set of 10,000 female UKBB participants are plotted against their age. (**C**) Number of age-sensitive phenotypes plotted against the declining number of people in whom these phenotypes were measured. (**D, E**) Sex hormone binding globulin concentrations (UKBB field# 30830) of a random set of 10,000 males (**D**) and females (**E**). (**F**) Average number of lifetime sexual partners is plotted against the age of UKBB participants (UKBB field ID# 2149). Gray area denotes 99% confidence interval. Color of dots on the plot represents relative density of dots in the area.

(see *Supplementary file 3* and *Supplementary file 4*). Examples include systolic blood pressure (shown in *Figure 1*), which increases with age, and hand-grip strength (shown in *Figure 1B*), which decreases with age.

Most large human databases and datasets, including UKBB, have certain limitations, such as incomplete or missing data points. Therefore, before proceeding to modeling aging, we needed to address the following three issues:

1. Certain phenotypes, such as MRI brain scans, were only available for a subset of UKBB participants (in this case <50,000). Therefore, we could not use MRI data to estimate the age of the remaining participants. Thus, the inclusion of such incomplete phenotypes in the UKBB database required an optimization strategy. The objective was to identify individuals who appeared

young for their age, and the more individuals in the study, the greater the likelihood of discovering them. Likewise, including more diverse phenotypes improves the robustness and global assessment of overall aging. However, as we increased the number of age-dependent phenotypes, the number of individuals evaluated decreased. From the curve's shape (*Figure 1C*), we estimated an optimal inclusion threshold to be ~120±15 phenotypes.

2.  Significant phenotypic differences exist between the sexes. For example, the parameters 'age at which first facial hair appeared', 'age at menopause', and 'degree of pattern balding' are gender-specific. Additionally, shared phenotypes may have different dynamics in males vs females. For example, increasing plasma concentration of sex hormone binding globulin is one of the best predictors of age in males (*Figure 1D*); however, in females, its plasma concentration stays nearly constant or even tends to decrease (*Figure 1E*). Thus, we analyzed male and female aging separately.

3.  The dataset we used is largely cross-sectional, meaning that each data point represents a different person at a different age. Consequently, phenotypes that are used to predict age could be indicative of cultural and societal changes over time, rather than biological changes associated with aging. For instance, a good predictor of age (with a p-value<$10^{-52}$) is the lifetime number of sexual partners (*Figure 1F*). While sexual activity and fertility have been linked to human aging and longevity (*Min et al., 2012*), the correlation here is most likely driven by evolving social norms in Britain. Other examples of such traits include 'how many siblings do you have' or 'how long have you lived in your current house.' Moreover, some biological measurements were derived using age as a parameter. For instance, basal metabolic rate (BMR) is an outstanding age predictor (p-value<$10^{-255}$). However, BMR was not measured directly; instead, it was computed using a formula that incorporates height, weight, gender, and age itself. Therefore, we examined each age-dependent parameter independently, aiming to satisfy three broad criteria: (a) the trait should not reflect societal norms and structures; (b) the trait should not be a function of elapsed time (e.g. how long have you been drinking green tea?); and (c) the trait's value should not depend on a person's actual age. We endeavored to use purely biological and physiological parameters. Although it is possible that the selected phenotypes were still influenced to some degree by the birth cohort, these considerations should have reduced this effect. The complete list of age-related traits we selected, along with the reasons behind our choices, can be found in *Supplementary file 3 and 4*.

## Age-dependent physiological traits fall into clusters

The phenotypes we selected for our age prediction model were often correlated to one another, e.g., left-hand and right-hand grip strength. To assess the degree and pattern of correlations among the age-dependent traits (see *Supplementary file 1*), we first normalized each phenotype by its mean and standard deviation. For phenotypes represented as multiple-choice questions (e.g. do you take naps - often, sometimes, rarely, never?), we encoded each answer option as a binary vector (one or zero), and these vectors were also normalized. Correlations were computed for each pair of phenotypes and visualized as dendrograms (*Figure 2A and B*). As expected, highly correlated phenotypes grouped together, such as 'BMI'-'weight'-'waist circumference' or 'cholesterol'-'LDL'. Surprisingly, this analysis uncovered strong correlations that were not obvious, such as 'I drive faster than the speed limit most of the time (id# 1100)' with 'I like my drinks very hot (id# 1518)' (*Figure 2A and B*; marked with yellow shadows). Notably, most of the clusters appeared to be enriched for phenotypes associated with a specific organ or physiological system. For example, the cluster that contains 'creatinine', 'urea', 'cystatin-C', and 'phosphate' likely reflects kidney function, whereas the cluster that contains 'systolic blood pressure' and 'diastolic blood pressure' likely reflects cardiovascular function (*Figure 2A and B*). That said, upon close examination, it is not intuitively obvious why some physiological traits do or do not cluster with one another. Thus, this dendrogram might be a valuable data source for future hypothesis generation and exploration.

## A mathematical model to predict age

To develop a model that predicts age, we experimented with several algorithms, including simple linear regression, gradient boosting machine (GBM), and partial least squares (PLS) regression. Different approaches have different advantages and limitations; however, we decided to choose one approach and not develop and analyze several independent models in parallel in order to not artificially inflate the false discovery rate (FDR). We ultimately selected PLS regression because it enabled

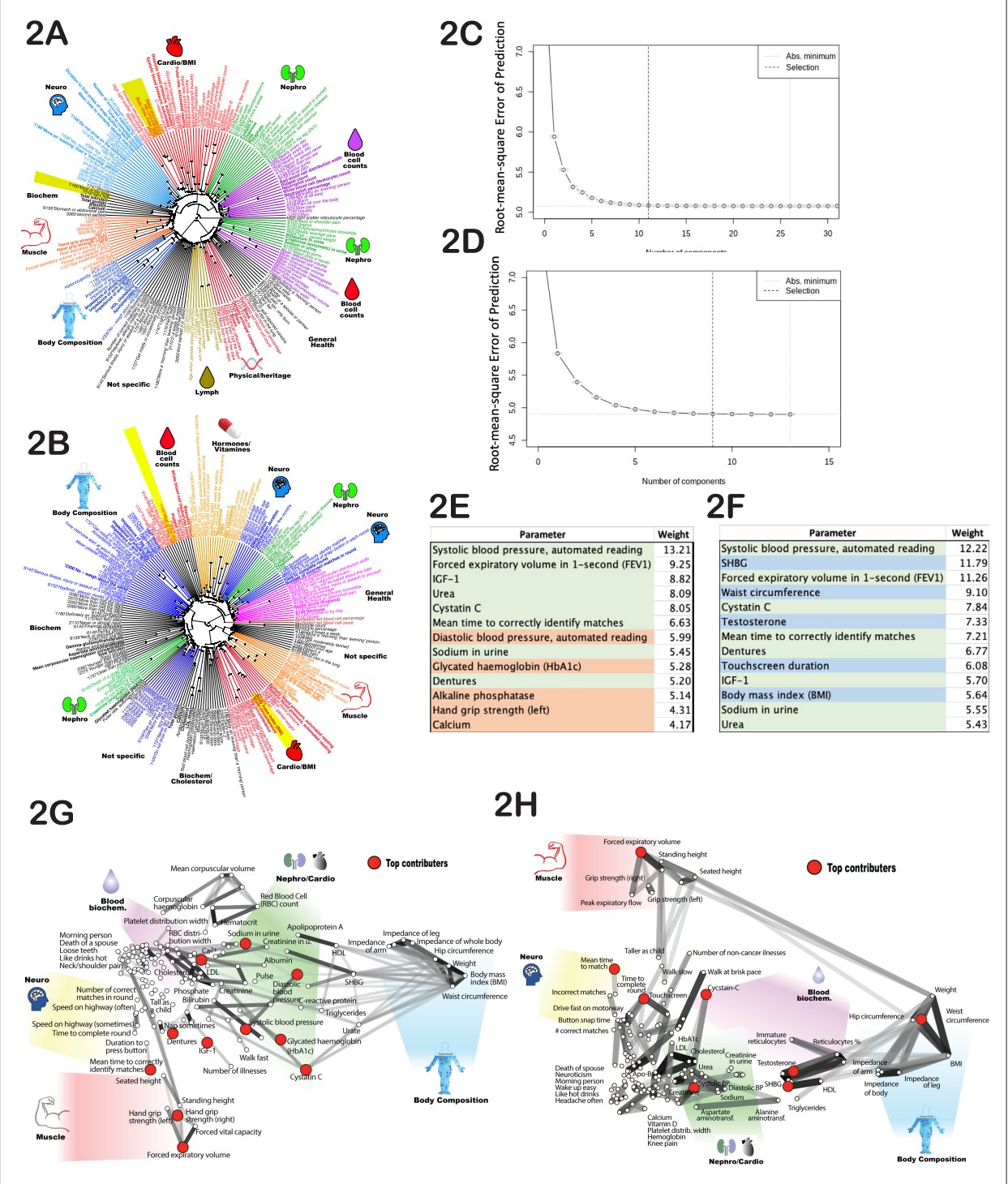

**Figure 2.** Age-dependent phenotypic clustering. Dendrogram plots of age-dependent female (**A**) and male (**B**) phenotypes selected for age prediction. Numbers in the name of 'rays' represent United Kingdom BioBank (UKBB) ID numbers for multiple-choice questions (see **Supplementary file 1** or the UKBB website), followed by the answer. Major clusters were colored and subjectively assigned a name that reflects a possible biological function of the cluster. The number of principal components included in the PLS (projection to latent structures) model to predict age vs root mean square error of the predictions is plotted for females (**C**) and (**D**) males. (**E**) The top phenotypes with the highest weights in the age-predicting PLS model are listed for (**E**) females and (**F**) males. Phenotypes shaded in green are shared between sexes, red are specific to females, and blue are specific to males. All phenotypes were used for both sexes, and this shading reflects only the position in the list of top 13 traits. (**G**) List of phenotypes used to predict age of females and (**H**) males projected on 2D space using correlation as the distance measure. The degree of correlation is also depicted by gray lines. The darker the shade, the stronger the correlation. Note that the distortion in positioning is an inevitable consequence of projecting high-dimensional data

*Figure 2 continued on next page*

*Figure 2 continued*

into 2D space. As before, groups of related phenotypes were subjectively assigned a name that likely depicts their physiology, and phenotypes with the highest weight in the PLS model were depicted by red dots.

us to determine the number and composition of components required to predict age optimally from the data, which provides additional insights into the biology of human aging. But before making this selection, we compared the performance of the three approaches. The outcomes of PLS and linear regression were almost identical (R-squared between ΔAge values derived by these two methods was 0.99, meaning that if one model were to predict an individual was 62 years of age, the other model would have the same prediction). This similarity is likely due to the small number of predictors (121 phenotypes) and comparatively large number of participants (over 400,000). The correlation between GBM model outcomes and PLS (and linear regression) was slightly smaller (R-squared=0.87). The reason for the lower correlation is likely the need for imputation in PLS and linear regression models. The GBM model tolerates missing data, whereas linear regression and PLS methods require imputation or removal of individuals with too many data points missing, an approach we describe in more detail below.

PLS modeling is not tolerant of missing values, and in the UKBB dataset we used, over 60,000 participants (~15%) lacked at least one phenotypic measurement. To prevent excessive imputation, we excluded any individual missing more than 15 data points from the study, thereby decreasing the number of selected female participants from 222,111 to 215,949 (~2.7% loss) and males from 188,609 to 183,715 (~2.6% loss). We imputed and scaled the values of the remaining participants with missing data (Methods).

Next, we determined how many PLS components (each derived from UKBB phenotypes) were required to predict chronological age. To do so, we constructed a series of age prediction models using an increasing number of these components. The first model was built using only component #1, the second using components #1 and #2, and so on. At each step, we calculated the root mean square error of the age prediction and determined its decline using the R function 'selectNcomp' (see *Figure 2C and D*). Our analysis revealed that only 11 independent components were required to describe female aging dynamics, and 9 independent components were required for males. Including additional components did not further improve the model performance. Therefore, we used the R function 'plsR' with 9 and 11 components for males and females, respectively, along with the cross-validation function (CV) to prevent overfitting when building models to predict age using UKBB phenotypes. Specifically, we performed 10 rounds of cross-validation, where 10% of data were held out and the remaining 90% used for training. Over 10 rounds, different 10% were held out for validation. In each case, the findings were validated in the test set. Full statistics and approach are described in Methods.

It was interesting to determine which individual age-sensitive phenotypes were most useful for age prediction. Since many phenotypes contribute to multiple PLS components, we deconstructed each PLS component and calculated the sum of the absolute values for phenotype coefficients across all components. This provided a weight metric for each phenotype used to predict age. The top 13 phenotypes with the highest weights are presented in *Figure 2E and F*. Most were shared between males and females and were associated with different physiological systems, e.g., systolic blood pressure (which likely correlates with cardiovascular health), forced expiratory volume (FEV) (pulmonary and cartilage/bone health), urea and cystatin-C levels (kidney health), and mean time to correctly identify matches (cognitive health). Moreover, if we deleted one of these selected traits, the model substituted a close correlate; specifically, it substituted 1 s FEV for FEV, systolic blood pressure for diastolic blood pressure, and hand-grip strength (right) for hand-grip strength (left). The fact that the model best predicted chronological age when it received input from a wide range of physiological systems underscores the global, systemic nature of the aging process. Similar conclusions were drawn from high-dimensional analysis of aging mice (*Chen et al., 2022*).

## Inferred (biological) age predicts all-cause mortality better than chronological age

We utilized the physiological phenotypes listed in *Supplementary file 3* and *Supplementary file 4* and the PLS modeling described above to predict female age with a root mean square error of

4.8 years, $R^2 \sim 0.63$, and predict male age with a root mean square error of 5.1 years, $R^2 \sim 0.6$. Several factors may contribute to discrepancies between predicted biological age and chronological age, including statistical noise, variations in life histories among UKBB participants, limited accuracy of certain measurements, and inadequate numbers of relevant measurements. However, some of this discrepancy may arise because certain individuals are aging more slowly or rapidly than the mean for that age. Consistent with this interpretation, we observed a significant correlation of residuals between two assessments for a small number of UKBB participants who were evaluated longitudinally (twice) with intervals of up to 12 years ($R^2 \sim 0.56$, $p < 10^{-255}$).

To estimate biological age from this cross-sectional data, we computed a value termed ΔAge for each participant. We define ΔAge as the individual's chronological age subtracted from their predicted age and normalized such that the average ΔAge for the entire population at each age is zero. ΔAge is negative if an individual is predicted to be younger than they are and positive if an individual is predicted to be older. The ΔAge parameter carries no information about the person's actual chronological age, as it is equally distributed across zero at any age (*Figure 3A*). Comparable approaches have been employed previously, such as using DNA methylation patterns (*Marioni et al., 2015*) or facial images and computer vision (*Chen et al., 2015*) to predict age and identify potentially 'fast agers' and 'slow agers'.

## One year of ΔAge carries approximately the same mortality risk as 1 year of chronological age

The classical paradigm of aging described by Gompertz stipulates that mortality rates increase exponentially with time, doubling roughly every 8 years (*Kirkwood, 2015*). In the UKBB dataset that we analyzed, a small number of participants (8883 males and 5668 females, *Figure 3B*) passed away within 5 years of their initial test-center attendance. The distribution of these deaths among UKBB participants has a typical 'Gompertzian' shape, with mortality rates exponentially doubling every 7.7 years for both males and females (*Figure 3C*). In Gompertz' model, where mortality depends only on age, everyone of the same age has an equal likelihood of dying. However, by incorporating ΔAge, we were able to further forecast death among individuals of the same age. To illustrate this point, consider males who are 62 years of age and group them based on their ΔAge (as shown in *Figure 3D*). Individuals on the left side (with negative ΔAge values) were predicted by the model to be younger than 62, while those on the right were predicted to be older. In this UKBB sub-cohort, several hundred subjects died within 5 years following their enrollment. Plotting the average mortality for each ΔAge bin in this stratification of 62-year-olds resulted in a Gompertz-like mortality distribution (*Figure 3E*). Notably, the effect of 1 year of ΔAge on the mortality rate was almost identical to that of 1 year of chronological age. It is important to emphasize that death data were not considered during the development of the model of biological age or derivation of ΔAge, and that ΔAge does not exhibit any correlation with chronological age (as illustrated in *Figure 3A*). The ability of ΔAge to predict mortality has a similar level of accuracy as chronological age. This ability is consistent across genders and ages and can even be observed when individuals of all ages are combined (*Figure 3F*). We consider this progressive increase in mortality rates with progressively larger ΔAge to be a powerful validation of this modeling strategy for assessing biological age. The fact that combining chronological age with ΔAge leads to a more precise prediction of mortality risk than relying on chronological age alone might be of interest to actuaries.

## ΔAge correlates with parental lifespan

Remarkably, we observed a robust correlation of ΔAge with the age at death of the participant's father (p-value=$1.9*10^{-43}$ for females and p-value=$3.9*10^{-31}$ for males) and mother (p-value=$1.1*10^{-68}$ for females and p-value=$1.3*10^{-32}$ for males). Individuals predicted to be biologically younger had parents who lived longer. Previous studies have reported that the lifespans of parents and offspring are correlated (*Ruby et al., 2018*; *Milman and Barzilai, 2015*). These findings, too, provide strong validation for the model, reinforcing the idea that ΔAge is not simply noise, but rather carries significant information about the aging process and its variability in the population.

## Environmental factors that influence biological age

Previous studies have shown that personal wealth is positively associated with human lifespan (*Chetty et al., 2016*; *Wang and Geng, 2019*), whereas smoking and excessive drinking are negatively

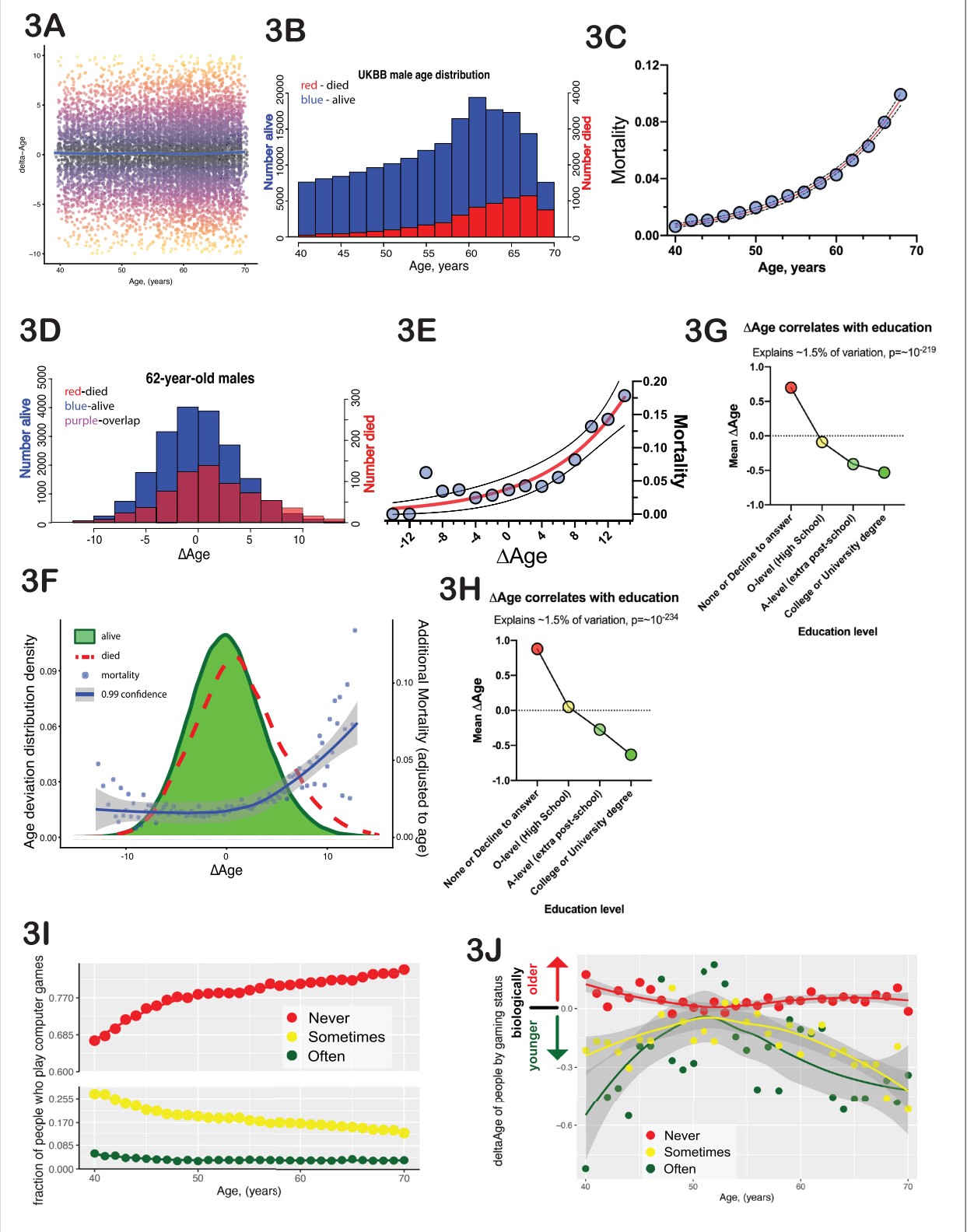

**Figure 3.** ΔAge has biological meaning. (**A**) Delta-age (ΔAge, predicted biological age minus chronological age) is plotted against chronological age for a random subset of 10,000 United Kingdom BioBank (UKBB) participants. Note that there is no correlation between age and ΔAge. (**B**) Histogram of age distribution (blue) and death distribution (red, right y-axis) is presented for UKBB males. (**C**) Mortality of UKBB male participants vs their age is plotted; note the classical exponential (Gompertzian) shape. Blue dots are actual data, the red line is an exponential fit, and the black dash line is 95%

*Figure 3 continued on next page*

*Figure 3 continued*

confidence interval. (**D**) Histogram of the ΔAge distribution (blue) and death distribution (red, right y-axis) is presented for UKBB males of 62 years of age only. (**E**) Mortality of 62-year-old males is plotted against their ΔAge. Blue dots are actual data, the red line is an exponential fit, and the black dashed line is 95% confidence interval. Once again, note the classical exponential (Gompertzian) shape with ΔAge, even though all the subjects are the same age chronologically. (**F**) Distribution of ΔAge for all the people in UKBB (all ages and all genders, green shape). The distribution of ΔAge for people who died within 5 years after enrolling in the UKBB (red line) is shown for comparison; note a shift of the deceased distribution to the right toward larger ΔAge (predicted older on average). The mortality penalty due to ΔAge is plotted as blue dots (left y-axis), the exponential fit of these data is presented as a blue line, and the 99% confidence interval as a gray shade. (**G**) Average ΔAge is plotted for UKBB males (**G**) and females (**H**) against their highest education (qualification) level achieved. (**I**) The fraction of people who play computer games 'sometimes' (yellow dots), never (red dots), and people who play computer games 'often' (green dots). (**J**) Average ΔAge of people at different ages separated by their computer gaming habits (see I). As a group, people who play computer games 'often' are biologically younger than people who play computer games 'sometimes' or 'never'.

The online version of this article includes the following figure supplement(s) for figure 3:

**Figure supplement 1.** Men and older women who play computer games 'often' on average are more youthful.

associated with lifespan. To investigate whether this measure of physiological ΔAge has similar associations, and possibly to identify new environmental factors that influence aging, we calculated the correlation of ΔAge with every parameter available from UKBB (*Supplementary file 5* and *Supplementary file 6*). Correlations with p-values lower than $10^{-5}$ (calculated to correct for multiple testing) were considered statistically significant. Interestingly, we observed a strong association of ΔAge with age-dependent biological phenotypes that were not included in the model to predict ΔAge due to the low number of people who underwent the assessment. For example, heel bone density (UKBB field #3148) and thalamus volume (UKBB field #25011) both had strong associations with ΔAge (p-values were ~$10^{-11}$ and $10^{-10}$, respectively). These and other phenotypes with strong ΔAge correlations again help to validate the model and might be useful parameters to consider when building biological clocks in the future.

Tables in *Supplementary file 7* and *Supplementary file 8* list the environmental factors we found to correlate with ΔAge. As predicted, wealth was positively correlated with a more youthful ΔAge. For instance, parameters such as 'home location' (UKBB field id# 20075), 'place of birth' (UKBB field id# 129), 'Townsend deprivation index' (UKBB field id# 189), and 'total income' have a strong and significant correlation with ΔAge (*Supplementary file 5*, *Supplementary file 6*, *Supplementary file 7*, *Supplementary file 8*). Additionally, smoking and exposure to smoke (UKBB field ids# 20161 and 20162) were positively correlated with an older ΔAge. The impact of moderate alcohol drinking on long-term health is still a subject of debate. In our data, the overall frequency of alcohol consumption (numerous UKBB fields, like 20414) did not have a significant correlation with ΔAge; however, the alcohol type did. Consuming beer and hard cider (UKBB field id# 1588) was positively correlated with ΔAge, whereas consuming Champagne and other white wines (UKBB field id# 4418) was negatively correlated. It is likely that drinking Champagne frequently is an indicator of higher socioeconomic status.

The single most significant non-biological parameter that correlated with ΔAge in both males and females (p-value<$10^{-200}$) was 'qualifications' or the level of education achieved (UKBB field id# 6138). Each additional level of education was progressively associated with increased 'youthfulness' (*Figure 3G and H*). Interestingly, the effect size of education (–1.51) was much greater than that of wealth (–0.81) or place of birth (–0.13).

Certain leisure and social activities were also correlated with ΔAge. The amount of TV watching (UKBB filed# 1070) was positively correlated with ΔAge in both males and females, whereas time spent outdoors (UKBB filed# 1050) for males and DIY projects (UKBB filed# 2624) for females were correlated with younger ΔAge. Intriguingly, the second strongest behavioral trait that was associated with ΔAge was the 'frequency with which people play computer games'. This is a novel association, and one that is less likely to reflect socioeconomic status, as access to computer gaming is inexpensive and widely available. Playing computer games associated with youthfulness (*Figure 3I and J*, *Figure 3—figure supplement 1*), with a size effect of –2.2 and p-value of $4*10^{-8}$. This association was equally strong if 'age' was factored out from the regression, indicating that generational changes in leisure activities do not explain this association.

## Genetic loci associated with biological age

To identify potential genetic determinants of physiological ΔAge, we carried out a genome-wide association study (GWAS), using linear models separately on males and females (Methods). Manhattan plots for male and female GWAS models are presented in *Figure 4A–D* (summary statistics is deposited to and available from https://www.ebi.ac.uk/gwas/, accession numbers: GCST90566392 [for females] and GCST90566393 [for males]). The inflation factor in our analysis was $\lambda_{gc}$=1.2005 for males and $\lambda_{gc}$ = 1.2531 for females. Linkage disequilibrium (LD) regression intercepts were 1.0213±0.0083 and 1.0285±0.0119 for males and females, respectively.

Using a stringent multiple testing correction for GWAS (*Chen et al., 2021*) with a threshold of $10^{-9}$, we identified 9 loci associated with ΔAge in males and 25 loci in females (*Figure 4A and B*). Four of these loci were found in both sexes. Specifically, these include the HLA locus, located at chr6:32,600,000; chr10:64,900,000, a locus that contains NRBF2, JMJD1C, and TATDN1P1 genes; chr19:45,413,233, a locus that contains APOE, TOMM40, and APOC genes; and chr20:23,613,000, a locus that contains the CST3 gene. These genes are strong candidates to influence whether a person is biologically young or old for their age. Two of these loci, APOE (*Schächter et al., 1994*; *Sebastiani et al., 2019*) and HLA (*Yang et al., 2017*), have previously been associated with human longevity, which increases our confidence in the analysis. GWAS analysis of combined male and female ΔAge data identified 12 additional loci (and candidate genes associated with these loci), which are listed in *Figure 5B*.

Additionally, we compared our ΔAge GWAS association results with similar GWASs that were performed for other biological clocks. For example, *McCartney et al., 2021*, used DNA methylation data on 40,000 individuals to compute biological age called GrimAge. After that, they calculated an intrinsic epigenetic age acceleration (a value similar to ΔAge, which measured a deviation of biological age from chronological age) and performed GWAS.

## A healthy sub-cohort distinguishes genes that affect aging vs age-related disease

Some genes that are associated with ΔAge in our analysis are known disease risk factors. For example, the HNF1A (hepatocyte nuclear factor 1 homeobox A) locus (top SNP - rs1169284, ΔAge association p-value=$3.0*10^{-23}$) is associated with diabetes (*Shepherd et al., 2009*) and cancer (*Abel et al., 2018*). The APOE (apolipoprotein E) locus (top SNP - rs7412, ΔAge association p-value=$4.4*10^{-33}$) is associated with Alzheimer's disease and coronary heart disease (*Xu et al., 2016*).

It is possible that people who carry risk alleles for age-related disease have a higher ΔAge due to the disease itself, even though their aging may be unaffected otherwise. To investigate this, we calculated the association of top loci with ΔAge in a 'healthy-only' cohort, excluding people who had been diagnosed with disease, specifically, diabetes, cancer, asthma, emphysema, bronchitis, chronic obstructive pulmonary disease, cystic fibrosis, sarcoidosis, pulmonary fibrosis, tuberculosis, any vascular or heart problems (such as high blood pressure, stroke, angina, or heart attack) or anybody with a history of allergic complications. These exclusion criteria decreased the number of people in the study by almost 50%; however, the association of ΔAge for top hits remained. These findings suggest that most of the genetic signal associated with ΔAge comes not from a few susceptibility alleles for specific diseases but rather from alleles that describe and possibly drive fundamental processes that change with age, i.e., possibly with aging itself. Conversely, this analysis also identified genes that were specifically responsible for certain diseases that present similarly to accelerated aging. For instance, the GCKR (glucokinase regulatory protein) locus showed a strong association with ΔAge (p-value=$8*10^{-12}$); however, the association disappeared when we excluded individuals diagnosed with diabetes. This demonstrates that mutations in GCKR cause a disease that resembles aging but do not have a detectable effect on the overall aging of healthy individuals.

Nonetheless, caution should be exercised when interpreting the analysis of this smaller, 'healthier' subpopulation. It is possible that certain hits disappeared not due to disease but because of decreased statistical power resulting in false negatives. Conversely, some individuals may have had undiagnosed or subclinical disease, leading to false positives. Additionally, some of the associations may be false positives due to collider bias. Thus, we favor the interpretation that among the GWAS hits that disappeared in the healthy sub-cohort were disease-susceptibility genes, while those that persisted likely

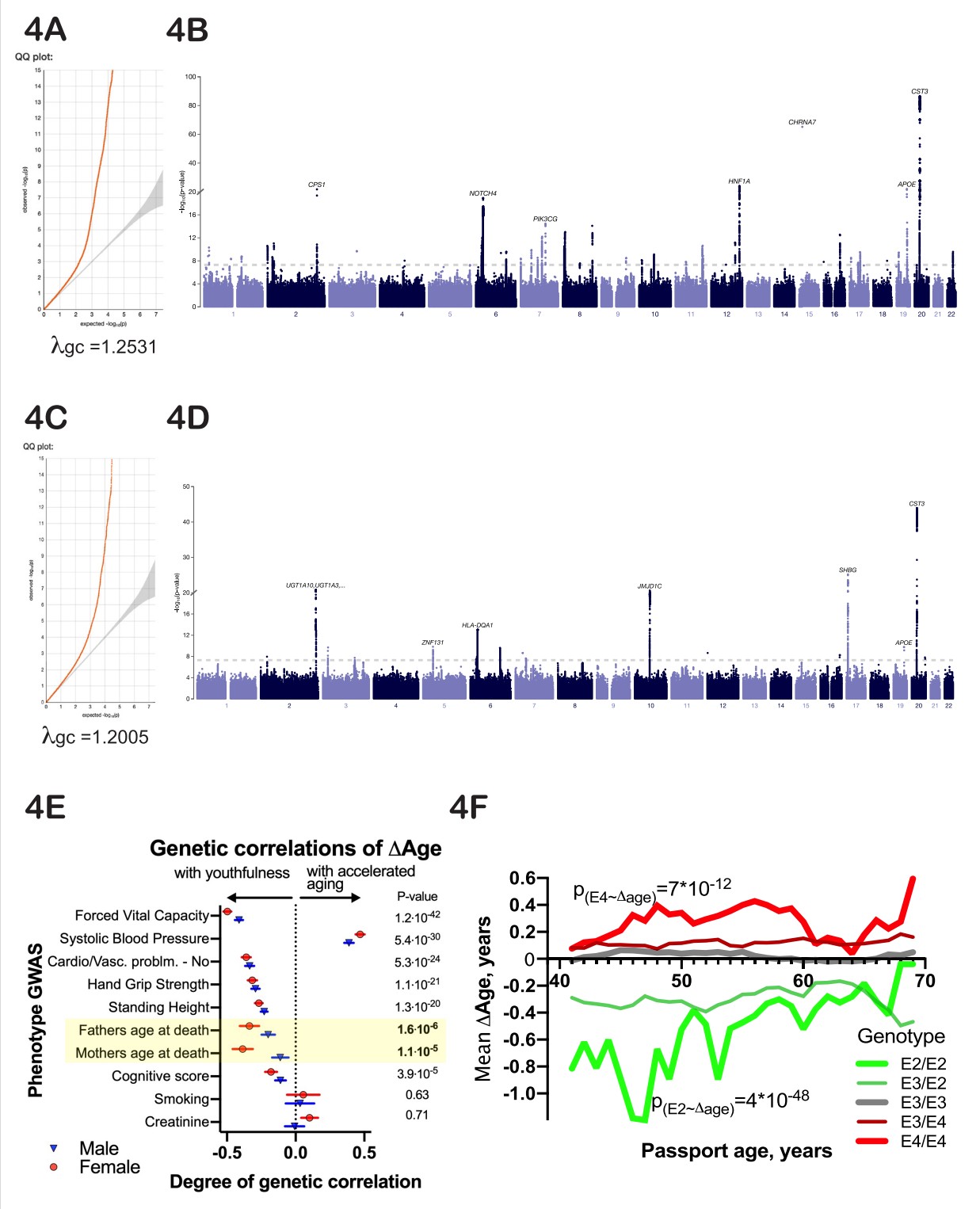

**Figure 4.** Genetic analysis of ΔAge. (**A, C**) Quantile-quantile plots for female and male -log10 p-values. (**B, D**) Manhattan plots from genome-wide association analysis of female and male ΔAge. (**E**) Correlation of ΔAge genome-wide association study (GWAS) determination with other GWAS performed and reported by the United Kingdom BioBank (UKBB) consortium. Note the strong genetic relation between GWASs for ΔAge and parental age at death. (**F**) Effect of *APOE* alleles on average ΔAge plotted across different ages. Beneficial *APOE2* alleles are in green, and detrimental *APOE4* alleles are in red.

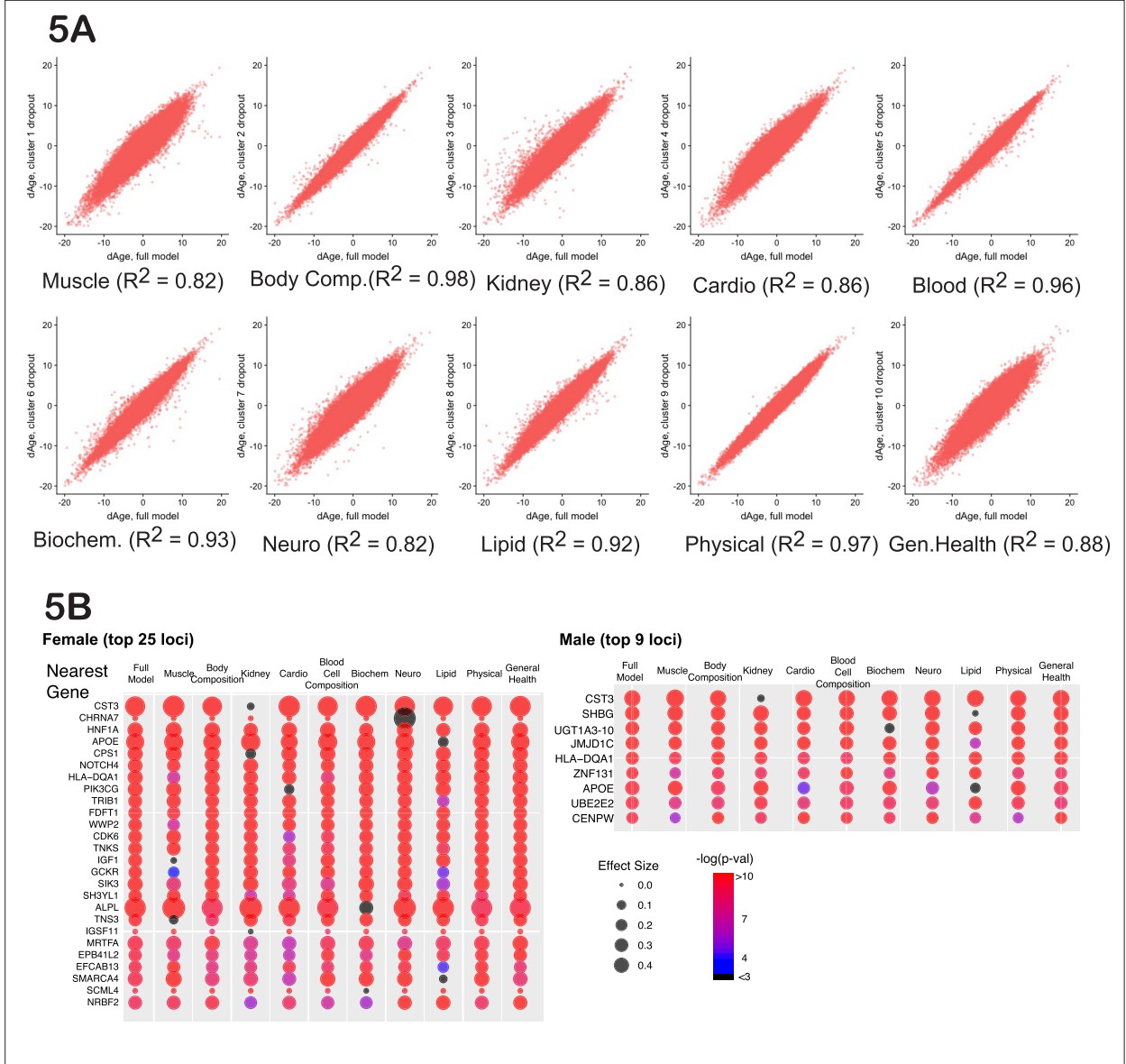

**Figure 5.** Cluster-dropout models. (**A**) Correlation between ΔAge calculated using full set of identified parameters and each of 10 dropout models. Note that ΔAge values remain robust between models, meaning that if the person is predicted to have large ΔAge by the complete model, the 'dropout' models will predict large ΔAge as well. (**B**) The list of genes nearest to genome-wide association study (GWAS) loci that associate with female and male ΔAge in the full model. Each hit is presented as a bubble, colored according to the significance of association of the locus with ΔAge, with size representing the effect size of the top SNP in the locus. The full summary statistics is deposited to and available from https://www.ebi.ac.uk/gwas/, accession numbers: GCST90566392 (for females) and GCST90566393 (for males).

influence the aging process more generally. Future longitudinal and other studies in humans and potentially animals could lend support to this interpretation.

## Heritability of ΔAge

To estimate heritability, we performed LD score regression analysis (*Zheng et al., 2017*). The analysis involved 1,293,150 unique SNPs with an allele frequency higher than 0.01. We found that total genetic heritability ($H^2$) of ΔAge was ~11% (0.108±0.009) for females and ~10% (0.096±0.008) for males, which is similar to the genetic heritability estimated previously for human longevity (*Ruby et al., 2018*; *Melzer et al., 2020*). This may be because the variation in genetic diversity is not substantial or because existing alleles of critical longevity genes do not have significant effect sizes in this human population.

## GWAS signatures that correlate with the ΔAge GWAS

Another way to infer the biological meaning of ΔAge is to compare the GWAS signatures (Manhattan plots) of ΔAge to GWAS signatures of other traits in public databases (*Zheng et al., 2017*). We found that the genetic signatures of some of the components used to calculate ΔAge were correlated with the genetic signature of ΔAge itself (*Figure 4E*). For example, GWAS of forced vital capacity (FVC) had a correlation with ΔAge GWAS of 0.49±0.02 (p-value=5*10$^{-65}$). In fact, remarkably, the most similar GWASs together spanned multiple organ systems (pulmonary, cardiovascular, musculature, cognition), arguing that this 'aging' GWAS integrates the health of multiple organ systems.

In contrast, GWAS signatures of certain physiological parameters, such as blood creatinine levels, which were explicitly used in ΔAge derivation, had no genetic correlation with ΔAge (0.1±0.07, p-value=0.1). It is possible that traits whose GWAS signatures genetically correlate with the GWAS signature of ΔAge are drivers of aging, while traits with uncorrelated GWAS signatures are simply biomarkers. Certain metabolic parameters have been correlated with mortality in previous studies (*Deelen et al., 2019*), but it has been an open question if those metabolites have a causal relationship to aging and mortality.

It is interesting to note that the genetic signature of ΔAge has a strong similarity to the genetic signature obtained through GWAS for 'father's age at death' and 'mother's age at death' (*Figure 4E*). This correlation was present even though the mortalities of subjects or parents were not part of the model and were not considered throughout the analysis. The genetic correlation of GWAS for parent's age mortality with GWAS for offspring's ΔAge was 0.39±0.03, p-value=1*10$^{-7}$ for females and 0.2±0.05, p-value=3*10$^{-5}$ for males. These GWAS correlations further demonstrate that ΔAge carries information about aging and longevity, despite its values being derived from cross-sectional physiological data and being independent of lifespan.

## GO highlights a neuronal influence on biological age

To investigate whether specific pathways or systems have an influence on biological age, we performed gene ontology (GO) analysis of extended GWAS hits (combined male and female genetic loci identified by the closest ORF). Five enriched pathways were identified in this analysis (*Table 1*). Unexpectedly, the top enriched category (GO:98815) was *modulation of excitatory postsynaptic potential*, enriched ~18-fold over the expected by-chance reference, with a multiple-testing-adjusted p-value of 0.046. This category was exceptional (~18-fold enrichment), as the second-best enrichment category was enriched only ~3-fold (*response to oxygen-containing compounds*). This GO category comprised multiple genes influencing synaptic function (*Table 1*), suggesting that the nervous system plays a particularly important role in aging systemically. Like the vasculature, the sympathetic nervous system impacts the function of many peripheral organs, and synapse function plays a critical role in the function and the maintenance of the CNS. Hints of such an association have come from genetics studies of

**Table 1.** Top gene ontology (GO) association of ΔAge genome-wide association study (GWAS) hits performed using an ontology resource previously described (*Mi et al., 2019*) using an online engine available at http://geneontology.org/ identifies synaptic category as the most overrepresented.

| GO biological process complete | Homo sapiens - genome | GWAS hits | Fold enrichment | Raw p-value | Corrected p-value |
|---|---|---|---|---|---|
| Modulation of excitatory postsynaptic potential (GO:0098815) | 43 | 5 | 17.74 | 1.49E-05 | 4.68E-02 |
| Regulation of cellular process (GO:0050794) | 11067 | 98 | 1.35 | 8.47E-06 | 6.67E-02 |
| Regulation of biological process (GO:0050789) | 11522 | 100 | 1.32 | 1.70E-05 | 4.47E-02 |
| Response to oxygen-containing compound (GO:1901700) | 1566 | 28 | 2.73 | 1.14E-06 | 1.79E-02 |
| Negative regulation of cellular metabolic process (GO:0031324) | 2545 | 35 | 2.1 | 1.82E-05 | 4.10E-02 |
| Cell communication (GO:0007154) | 5160 | 57 | 1.69 | 1.46E-05 | 7.68E-02 |
| Cellular response to stimulus (GO:0051716) | 6376 | 66 | 1.58 | 1.48E-05 | 5.84E-02 |

worms (*Apfeld and Kenyon, 1999*; *Li et al., 2016*), flies (*Libert et al., 2007*), and laboratory rodents (*Garratt et al., 2022*; *Zullo et al., 2019*).

GO association was performed using an ontology resource previously described (*Mi et al., 2019*) using an online engine available at http://geneontology.org/.

## Cluster-dropout analysis enriches for GWAS hits that influence aging globally

If a GWAS hit influences aging itself, reflecting the function of all the organs and physiological systems, the association between the SNP and ΔAge should not disappear if any one measurement is omitted from the model. Thus, we investigated the robustness of the GWAS hits in a systematic way, using what we term 'cluster-dropout models'. Specifically, we constructed a series of male and female models to predict ΔAge by systematically excluding small sets of highly correlated phenotypic clusters. We built 10 models, in which phenotypic clusters related to muscle (dropout model 1), body composition (2), kidney health (3), cardio health (4), blood cell composition (5), blood biochemistry (6), neuropsychiatric phenotypes (7), lipid metabolism (8), physical attributes (9), or general health (10) were excluded. The list of phenotypes belonging to each cluster is reported in *Supplementary file 3* and *Supplementary file 4* (for females and males) and was guided by the clustering presented in *Figure 2*. As expected, the ΔAge values remained consistent among all the dropout models (*Figure 5A*). This means that if a person was predicted to be ~x years younger or older than their chronological age, this prediction was approximately the same regardless of the phenotypic clusters omitted.

A systematic evaluation of cluster-dropout models can suggest which of the genetic hits from our original full-model GWAS are likely to influence organismal aging and which are linked to a narrower phenotype. To perform this analysis, we took the best SNP from each candidate GWAS locus from the full male or female analysis (above) and tested its association with ΔAge computed using each of the 10 dropout models. The bubble plots in *Figure 5B* represent the effect size of each of these SNPs (via bubble size) and the associated p-value (via color).

As predicted, some GWAS hits disappeared in certain dropout models. A particularly informative gene was CST3. CST3 encodes cystatin-C, a metabolite whose concentration increases with age. Levels of cystatin-C are routinely used to evaluate kidney health, and it is proposed to be used as a marker in the human aging study 'TAME' (*Justice et al., 2018*). Elevated levels of this metabolite had been linked to elevated risk of cardiovascular disease (CVD) (*van der Laan et al., 2016*), risk of cancer (*Jones et al., 2017*), and neurodegeneration (*Kaur and Levy, 2012*). However, in a Mendelian randomization study (*van der Laan et al., 2016*), it was shown that while levels of cystatin-C predict CVD well, SNPs that robustly alter expression of cystatin-C do not associate with CVD.

In the full model, CST3 had the most significant association with ΔAge (effect size>0.4, p-value<$10^{-80}$) in both males and females, as represented by its large red bubble. This association remained significant in all the dropout models, except dropout number 3 (kidney health clusters), which contains the CST gene product, cystatin-C concentration, which was one of the UKBB phenotypes used to generate the model. When the kidney clusters were omitted, the size effect of the CST3 association decreased to less than 0.1, p-value~0.1, which is represented by the small black bubble. Likewise, if we calculated ΔAge using all the inputs in the full model except for 'cystatin-C levels', the *CST3* locus was no longer associated with ΔAge. Combined, these data suggest that cystatin-C is a 'marker' rather than a driver or determinant of aging. In contrast, some GWAS hits never dropped out, and these remained candidates for fundamental determinants of physiological ΔAge.

To definitively distinguish whether a gene is a driver or a marker of aging, an experiment would need to be performed. It is possible that certain gene activities are influenced by existing FDA-approved medications, and retrospective analyses of human cohorts who take certain medications can be performed. More likely, however, an animal model would need to be employed, where animals with candidate genes modified via genetic means are investigated for lifespan and onset and progression of age-associated conditions. For example, one can engineer a mouse with a conditional allele of cystatin-C and evaluate how changes in dosage of this protein influence various phenotypes of aging.

In the same way, cluster-dropout models can be used to interrogate environmental factors. For example, as described above, computer gaming correlates with a youthful biological age (*Figure 3I and J*, *Figure 3—figure supplement 1*). The natural question is: are there specific physiological phenotypes, such as stimulus-reaction time or pattern recognition, that drive this correlation, or is it

reflective of a 'whole-body' biological age? To answer this question specifically, as well as to investigate all the phenotypes systematically, we calculated the strength of the correlation between every UKBB phenotype and all the cluster-dropout models (*Figure 5A*) in both males and females (presented in *Supplementary file 9*). To account for multiple testing, the Bonferroni corrected threshold of significance was $7*10^{-7}$. The correlation between biological age and computer gaming remained significant across all the models tested in both males and females, suggesting that there are no specific singular phenotypes responsible for this correlation. Such robustness of association was true for most phenotypes, but not all. For example, particulate air pollution (pm10) is associated with older biological age (p-value=$1.6*10^{-9}$ for females); however, if the model omits the cluster containing lung parameters, such as FEV, the correlation drops below Bonferroni-corrected statistical significance (p-value=$5*10^{-3}$ for females). This might suggest that particulate pollution mostly affects pulmonary health and, to a lesser extent, global organismal aging.

Another interesting observation is that the degree to which certain clusters contribute to the model does not necessarily correlate with how much this cluster contributes to genetic signature of human aging. For example, while dropping out cluster 10 (general health) resulted in quite significant changes of ΔAge distribution ($R^2$=0.88), the genetic signature as determined by GWAS did not change substantially. The most likely explanation is that many parameters in this category are influenced by environment more strongly than by genetics, e.g., not as much as caused by cluster 1 (muscle-related) removal.

One must keep in mind the caveats and complexity of comparing correlations of different phenotypes to each other, yet this dataset provides a good starting point for possible investigations of environmental factors influencing human aging.

## Discussion
### General caveats
Our study has several caveats. We used a cross-sectional dataset, where different ages are presented by people born at different times. Therefore, there is likely a 'cohort effect' in some or all predictors we use. Additionally, our model assumes that the rate of aging is constant for each individual, which is not always true. For example, a person's aging rate may change if they stop smoking. Despite these modeling assumptions, we believe that the final results are valid and generalizable and allow us to suggest new methods to measure physiological aging in humans and identify new targets to slow down human aging. The robustness of our modeling can also be assessed by considering a small number of UKBB participants (~13,000 out of ~500,000), who have been assessed twice, with the follow-up intervals ranging from 4 to 12 years. We observed a significant correlation ($R^2$~0.6, p-value$<10^{-255}$) between biological-chronological age measures for these individuals between their two assessments. This suggests that variation due to noise is not large. We also found that there is a significant correlation between longitudinally calculated rates of aging (change in biological age divided by assessment interval) and the values calculated using cross-sectional approach. Furthermore, to minimize the cohort effect in our genetic analysis, we used the year of birth as a covariate. Together with the correlations we observed between ΔAge and mortality, parent's mortality, previous GWAS longevity hits, and GWAS Manhattan plot comparisons, these findings suggest that the method we describe is a feasible approach to measure an individual's rate of aging and to identify genetic and environmental factors that may influence it.

### Broader implications of the model for physiological aging
How a general term like 'aging' maps onto age-dependent physiological traits is a deep question that may never be answered with great precision. In general, biological clocks can be used to identify new genes and environmental factors that influence aging, as we did here using this physiological clock. In addition, one can 'look into the clock' itself to gain additional insights. For example, we found that this mathematical model could best predict chronological age when all the different organ and physiological systems were sampled, emphasizing the systemic nature of aging. If the phenotypes associated with chronological age resulted from the decline of only one or a few organs, this would not be the case. Second, the model showed how different physiological traits covary and cluster in the population. Some correlations, such as vitamin D and sleep duration, are not immediately clear. However,

a post hoc examination of such an association can be explained in the light of previous medical research. We anticipate that exploring analogous nonintuitive clusters that cannot be explained currently may provide a new understanding of causal relationships. Third, the use of cluster-dropout models provided a powerful tool for distinguishing between individual genes and environmental factors that impact a specific physiological function from those that might affect all aspects of aging.

Many of the genes we identified are consistent hits in longevity GWAS analysis. Intuitively, this would be expected since aging is a risk factor for death. However, our model allows one to dig deeper and ask whether a longevity GWAS locus might be identified only because alleles prevent people from reaching an extremely old age. One could imagine that this is the case for *APOE*, since *APOE4* individuals generally die prematurely of Alzheimer's disease (*Olichney et al., 1997*; *Wright et al., 2019*). However, we find that at all ages, even as young as 40 years, *APOE* genotype influences ΔAge (*Figure 4F*), perhaps due to its more general effects on lipid homeostasis (*Abondio et al., 2019*) or inflammation.

To gain a deeper understanding of the genetic signature of ΔAge, it might be prudent to consider genetic loci that have a strong association with ΔAge (say p-value$<10^{-6}$), even though they do not reach the threshold for genome-wide statistical significance. While some of the loci in this expanded list can be false positives, many of the potential genetic determinants identified this way are of potential interest. The full summary statistics of the associations had been deposited to and is available from https://www.ebi.ac.uk/gwas/, accession numbers: GCST90566392 (for females) and GCST90566393 (for males).

To further analyze the meaning of genetic associations with ΔAge that we described above, we compared several published GWAS results obtained for human aging clocks using different health modalities. Specifically, we looked at GWAS for 'Epigenetic Blood Age Acceleration' (*Lu et al., 2018*), ML-imaging-based human retinal aging clock (*Ahadi et al., 2023*), PhenoAgeAcceleration and BioAgeAcceleration (*Kuo et al., 2021*), and the ΔAge GWAS we presented in our manuscript. Surprisingly, we discovered that there is no overlap between GWAS results for any two of these clocks built via different modalities - retina, DNA methylation, or physiological functions. However, there is a significant genetic overlap between clocks built using human phenotypic measures and our ΔAge model we describe. For example, the Biological Age Clock Acceleration calculated using HbA1c, albumin, cholesterol, FEV, urea nitrogen, SBP, and creatinine (*Levine, 2013*) in a US cohort (from National Health and Nutrition Examination Survey [NHANES]) yielded 16 significant hits in the GWAS analysis, 5 of which were also significant in our GWAS for UKBB-based ΔAge. These five common loci were close to the following genes: APOB, PIK3CG, TRIB1, SMARCA4, and APOE. The significance of this overlap is p$<10^{-8}$, suggesting that the ΔAge model we propose might be translatable to other cohorts of people.

An interesting question to consider is why GWAS results from other clock modalities, such as DNA methylation and retinal imaging, do not yield any genetic similarities to each other or to physiological and biological clocks. It is possible that these modalities of age assessment depend on completely genetically independent biological processes. For example, in a simplified manner, blood composition might be heavily weighted for DNA methylation, vascular structure for retinal scans, and muscle/bone/kidney health for physiological clocks. Data from model organisms suggest the master regulators of aging exist, and APOE is the best genetic variant known to influence human aging. Interestingly, only the biological and physiological clock models that we propose here pick it up as a hit. Alternatively, it is also possible that the true master regulators of aging rate are under stringent purifying selection, e.g., due to an important role in development, and, therefore, do not have genetic variability in human populations examined. As such, they could not be identified as hits in any GWASs.

When analyzing the many phenotypes that predict aging using PLS modeling, we discovered that only 9–11 axes are necessary to predict age. This suggests that there might be only ~10 independent systems (physiological networks) driving human aging. Interestingly, although overall the traits that figure most prominently into the sum of the principal components tend to map onto individual phenotypic clusters ('dropout clusters'), together the 'meaning' of the differentially weighted sets that comprise each principal component is not obvious. For example, the 10 dropout clusters we used are not representative of the 10 axes identified in PLS analysis. It would be interesting to understand the physiological significance of each axis to better understand the process of aging. That said, the two most valuable phenotypes used in our study (those that had, overall, the most weight in age

prediction) were FVC and blood pressure. Moreover, the genetic signature of ΔAge was similar to the genetic signature for FVC and blood pressure (*Figure 4E*). These phenotypes are integrated, multi-dimensional health measurements. Using genetic information to better understand the age-related phenotypes through PLS axis decomposition might be a fruitful direction for future research.

It is interesting to note that the three approaches we used to generate age prediction model (PLS, GBM, and linear regression) yielded very similar or identical results in performance. We chose to settle on one approach (PLS) to not artificially inflate the FDR; however, we verified that the top genetic loci associations obtained via the PLS model were also obtained in the GBM and linear models. Specifically, the top candidates (CST3, APOE, HLA locus, CPS1, PIK3CG, IGF1) identified in the PLS approach had statistically significant associations in all the models of ΔAge. It is likely that due to the small number of predictors (121) compared to a vastly larger number of individuals (over 400,000), the differences that these models introduce to final outcomes are quite small, which increases our confidence in the results.

Finally, from a practical perspective, we suggest that measuring human biological age using the 12 simple but diverse physiological measurements that together capture ~87% of the full ΔAge model (systolic blood pressure, FEV, and so forth; see *Figure 2*) might have actuarial and clinical value. For example, this physiological-age index could be measured longitudinally to learn how aging trajectories might be affected by environmental factors or antiaging therapeutics.

## Methods
### Phenotype data vector normalization
Before inclusion in the model or correlation analysis, each phenotype was first normalized to the mean and divided by its standard deviation. If the phenotype was encoded as a multiple-choice question (e.g. do you take naps - often, sometimes, rarely, never?), each answer option was encoded as binary vector (one or zero), and such vectors were normalized the same way. Euclidean distances and Pearson correlations were calculated for each pair of phenotypes.

### Correlation computation
For evaluation of the correlation between any two vectors, linear regression was performed using the 'lm' R function (stats package version 3.6.2). Adjusted $R^2$ and p-value had been extracted from such linear models using the 'summary(lm())' method. Where applicable, the significance threshold had been adjusted to account for multiple testing using Bonferroni correction.

Since PLS modeling and many other mathematical approaches do not tolerate missing values, and in the UKBB dataset that we selected, over 60,000 participants (~15%) lacked at least one measurement for one of the phenotypes, the following mitigation strategies had been used. To avoid excessive imputation, any individual missing more than 15 data points was excluded from the study. In females, the number of selected participants decreased from 222,111 to 215,949 (~2.7% loss), and in males from 188,609 to 183,715 (~2.6% loss). The rest had any missing data imputed using the R package 'BiocManager', function 'impute, version 3.8'. Imputation was performed using the 'KNN' algorithm.

PLS models were built in R using the package 'pls, version 2.7-2', function 'plsr'. All the phenotypes used in the modeling had been normalized to have a mean value of 0 and a standard deviation of 1, before being included in the model. This procedure ensured that the weight of phenotypes would not depend on the units of measurements. The number of components in the PLS model had been selected for each gender as described in the paper. Cross-validation was performed during the model generation with default parameters. Additionally, to test for overfitting, a PLS model had been generated on randomly selected 90% of individuals and tested on the remaining 10% with similar results, suggesting that no overfitting has occurred. An optimal number of components for the PLS model was identified as 11 for males with a minimum root mean square error (RMSE) achieved 5.1 years and 9 for females with a minimum root mean square error of 4.9 years. Cross-validation was performed using 10 segments and the R function 'crossval1'. In this implementation, dataset is split into 10 random segments, and 10 consecutive models are trained and tested. Each model is trained on 9 segments and tested on the tenth segment, such that each segment serves as a test segment once. Root mean square error of prediction is calculated in each instance for each combination of the components. The output of cross-validation is as follows:

For females:

Number of components considered: 9
VALIDATION: RMSEP
Cross-validated using 10 random segments.

|        | (Intercept) | 1 comps | 2 comps | 3 comps | 4 comps | 5 comps | 6 comps | 7 comps | 8 comps | 9 comps |
|--------|-------------|---------|---------|---------|---------|---------|---------|---------|---------|---------|
| CV     | 7.917       | 5.75    | 5.344   | 5.052   | 4.942   | 4.881   | 4.847   | 4.828   | 4.819   | 4.815   |
| adjCV  | 7.917       | 5.75    | 5.344   | 5.052   | 4.942   | 4.881   | 4.846   | 4.828   | 4.819   | 4.814   |

For males:

Number of components considered: 11
VALIDATION: RMSEP
Cross-validated using 10 random segments.

|        | (Intercept) | 1 comps | 2 comps | 3 comps | 4 comps | 5 comps | 6 comps | 7 comps | 8 comps | 9 comps | 10 comps | 11 comps |
|--------|-------------|---------|---------|---------|---------|---------|---------|---------|---------|---------|----------|----------|
| CV     | 8.086       | 5.959   | 5.341   | 5.341   | 5.271   | 5.202   | 5.164   | 5.142   | 5.130   | 5.121   | 5.114    | 5.109    |
| adjCV  | 8.086       | 5.959   | 5.542   | 5.340   | 5.271   | 5.202   | 5.163   | 5.141   | 5.129   | 5.120   | 5.113    | 5.108    |

AdjCV is bias-adjusted cross-validation of the error of prediction.

Where appropriate, boundaries of exponential fitting of the mortality data had been defined as 0.99 confidence. Mortality doubling time with age calculated from UKBB participants was very similar to that reported for the UK in other studies.

GWASs were performed using linear models separately on males and females. Analysis was performed using Hail-0.2.38 software, running on a Debian Linux cluster with 72 cores. To account for the influence of environmental factors on ΔAge, we corrected for smoking history, education, income level, Townsend deprivation index (a multiparametric measure of socioeconomic status), levels of air pollution, and geographic location of the primary house by including these variables as covariates. To account for and correct for the substructure of the population and possible batch effects, we included the first 40 principal components derived from genetic data of the participants and a genotyping batch number as covariates. Only people of white British ancestry were used for GWAS analysis. The GWAS results were analyzed using the PheWeb engine (version 1.1.14). The significance of GWAS hits was calculated considering LD and adjusted for multiple testing. The nearest genes for all identified loci were called using the GRCh37 human reference genome. Furthermore, we excluded questionable SNPs from the analysis due to irregularities in variant frequencies of certain SNPs identified in the UKBB dataset (*Kunert-Graf et al., 2020*), possibly due to their mis-mapping.

During dropout analysis, only the top SNP (single SNP in the associated locus with the smallest p-value) was considered. Both effect size (genetic beta) and nominal p-value were reported using the PheWeb engine.

LD score regression analysis of GWAS results and genetic correlation analysis were performed as described by *Zheng et al., 2017*, using Broad University online tool, specifically dedicated to this type of analysis. This analysis indicated that genetic heritability ($H^2$) of ΔAge was ~11% (0.108±0.009) for females and ~10% (0.096±0.008) for males. Additionally, for male ΔAge GWAS, $\lambda_{gc}$=1.2005 and LD regression intercept = 1.0213±0.0083. For female ΔAge GWAS, $\lambda_{gc}$ = 1.2531 and LD regression intercept = 1.0285±0.0119.

Specific details for data derivation (or data source) and processing (where applicable) are also described in the text and the corresponding figure captions.

## Acknowledgements

Authors are grateful to the Calico Community for support and discussions. Specifically, we are grateful to Eugene Melamud and his group for their support of the UKBB data framework, help with data processing and analysis, and numerous constructive discussions; we are grateful to Aarif Khakoo for his insights into disease-vs-aging paradigms; Kevin Wright and Graham Ruby for their discernment of human genetics of aging; and to David Botstein for his insights into statistical interpretation of our computational results and future applicability of such analysis. We are grateful to Madeleine Cule for supporting Calico's UKBB data interface and GWAS cluster maintenance and for guidance with GWAS methodology. We are grateful to Amoolya Singh for providing guidance in regression modeling, statistical analysis, and interpretation of the data. The authors are grateful to Jonathan K Pritchard for constructive discussions about modeling, genetic analysis, and results interpretation. This study was carried out using UK Biobank Application number 44584, and we thank the participants in the UK Biobank study. This study was funded by Calico Life Sciences LLC.

## Additional information

### Competing interests
Sergiy Libert, Alex Chekholko, Cynthia Kenyon: Works for Calico Life Sciences LLC, a pharmaceutical company engaged in understanding the biology of aging and development of therapies that ameliorate age-associated disease.

### Funding
No external funding was received for this work.

### Author contributions
Sergiy Libert, Conceptualization, Data curation, Formal analysis, Investigation, Visualization, Methodology, Writing – original draft, Project administration, Writing – review and editing; Alex Chekholko, Resources, Data curation, Software, Methodology; Cynthia Kenyon, Conceptualization, Resources, Supervision, Methodology, Project administration, Writing – review and editing

### Author ORCIDs
Sergiy Libert https://orcid.org/0000-0003-2497-5390
Cynthia Kenyon https://orcid.org/0000-0003-3446-2636

Reviewer #1 (Public review): https://doi.org/10.7554/eLife.92092.3.sa1
Author response https://doi.org/10.7554/eLife.92092.3.sa2

## Additional files

### Supplementary files
Supplementary file 1. Table showing the significance of associations between all available United Kingdom BioBank (UKBB) phenotypes and human data points with participant age in female subjects.

Supplementary file 2. Table showing the significance of associations between all available United Kingdom BioBank (UKBB) phenotypes and human data points with participant age in male subjects.

Supplementary file 3. Table showing a description of all the phenotypes considered for ΔAge model computation. This table includes the reasons for exclusion for each phenotype that was excluded as well as their assignment to correlation clusters for females.

Supplementary file 4. Table showing a description of all the phenotypes considered for ΔAge model computation. This table includes the reasons for exclusion for each phenotype that was excluded as well as their assignment to correlation clusters for males.

Supplementary file 5. Table showing the significance of associations between computed ΔAge (the normalized difference between chronological and calculated biological age) and all available United Kingdom BioBank (UKBB) phenotypes for females.

Supplementary file 6. Table showing the significance of associations between computed ΔAge (the normalized difference between chronological and calculated biological age) and all available United Kingdom BioBank (UKBB) phenotypes for males.

Supplementary file 7. Table showing the list of strongest associations between environmental factors and computed ΔAge (the normalized difference between chronological and calculated biological age). Additionally, all these associations are ranked by the size effect and the significance for all female participants.

Supplementary file 8. Table showing the list of strongest associations between environmental factors and computed ΔAge (the normalized difference between chronological and calculated biological age). Additionally, all these associations are ranked by the size effect and the significance for all male participants.

Supplementary file 9. Table summarizing the calculated strength of the correlation between every United Kingdom BioBank (UKBB) phenotype and all the cluster-dropout models (main *Figure 5A*) in both males and females.

MDAR checklist

## Data availability

This study analyzed a human dataset, which is publicly available through United Kingdom BioBank or UKBB (https://www.ukbiobank.ac.uk/; application number 44584). Calculations had been executed in publicly available software - R, Hail-0.2.38, and PheWeb, using functions described in the Methods. Access to dataset can be requested through https://www.ukbiobank.ac.uk/enable-your-research/apply-for-access. The code used to analyze the dataset is available at https://gitlab.com/biological-ageukbb/biologicalageUKBB-project/-/tree/master/public/R-scripts, copy archived at (*Libert, 2025*). GWAS datasets produced in a course of this study are deposited to and available from https://www.ebi.ac.uk/gwas/ under accession numbers: GCST90566392 (for females) and GCST90566393 (for males). The addresses for the accession are http://ftp.ebi.ac.uk/pub/databases/gwas/summary_statistics/GCST90566001-GCST90567000/GCST90566392/ and http://ftp.ebi.ac.uk/pub/databases/gwas/summary_statistics/GCST90566001-GCST90567000/GCST90566393/.

The following datasets were generated:

| Author(s) | Year | Dataset title | Dataset URL | Database and Identifier |
|---|---|---|---|---|
| Libert S, Chekholko A, Kenyon C | 2025 | Female delta-Age GWAS | https://www.ebi.ac.uk/gwas/studies/GCST90566392 | EBI GWAS Cataog, GCST90566392 |
| Libert S, Chekholko A, Kenyon C | 2025 | Male delta-Age GWAS | https://www.ebi.ac.uk/gwas/studies/GCST90566393 | EBI GWAS Cataog, GCST90566393 |

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
