## [Editor Report · eLife Assessment]

This **important** study developed a mathematical model to predict biological age by leveraging physiological traits across multiple organ systems. The results presented are **convincing**, utilizing comprehensive data-driven approaches. However, additional external validation could further strengthen its generalizability. The model provides a way to identify environmental and genetic factors impacting aging and lifespan, revealing new factors potentially affecting aging. It also shows promise for evaluating therapeutics aimed at prolonging a healthy lifespan.

---

## [Referee Report · Reviewer #1 (Public review)]

In this study, the authors developed a mathematical model to predict human biological ages using physiological traits. This model provides a way to identify environmental and genetic factors that impact aging and lifespan.

Strength:

(1) The topic addressed by the authors - human age predication using physiological traits - is an extremely interesting, important, and challenging question in the aging field. One of the biggest challenges is the lack of well-controlled data from a large number of humans. However, the authors took this challenge and tried their best to extract useful information from available data.

(2) Some of the findings can provide valuable guidelines for future experimental design for human and animal studies. For example, it was found that this mathematical model can best predict age when all different organ and physiological systems are sampled. This finding makes scenes in general, but can be, and have been, neglected when people use molecular markers to predict age. Most of those studies have used only one molecular trait or different traits from one tissue.

Weakness:

(1) As I mentioned above, the Biobank data used here are not designed for this current study, so there are many limitations for model development using these data, e.g., missing data points and irrelevant measurements for aging. This is a common caveat for human studies and has been discussed by the authors.

(2) There is no validation dataset to verify the proposed model. The authors suggested that human biological age can be predicted with a high accuracy using 12 simple physiological measurements. It will be super useful and convincing if another biobank dataset containing those 12 traits can be applied to the current model.

Comments on revisions:

In this revision, the authors improved the manuscript by adding discussion of two main weaknesses about human data limitation and model validation. My several other specific concerns and suggestions are all properly resolved.

---

## [Author Response]

The following is the authors’ response to the original reviews.

**Public Reviews:**

**Reviewer #1 (Public Review):**
Summary:In this study, the authors developed a mathematical model to predict human biological ages using physiological traits. This model provides a way to identify environmental and genetic factors that impact aging and lifespan.Strengths:(1) The topic addressed by the authors - human age predication using physiological traits - is an extremely interesting, important, and challenging question in the aging field. One of the biggest challenges is the lack of well-controlled data from a large number of humans. However, the authors took this challenge and tried their best to extract useful information from available data.

Authors thank an anonymous reviewer for agreeing that physiological clock building and analysis is an interesting and important even though challenging task.

(2) Some of the findings can provide valuable guidelines for future experimental design for human and animal studies. For example, it was found that this mathematical model can best predict age when all different organ and physiological systems are sampled. This finding makes sense in general but can be, and has been, neglected when people use molecular markers to predict age. Most of those studies have used only one molecular trait or different traits from one tissue.

Authors thank an anonymous reviewer for highlighting the importance of the approach we employ to sample traits for biological age prediction from multiple organs and systems, which ultimately provides more wholistic information

Weaknesses:(1) As I mentioned above, the Biobank data used here are not designed for this current study, so there are many limitations for model development using these data, e.g., missing data points and irrelevant measurements for aging. This is a common caveat for human studies and has been discussed by the authors.

Thank you for pointing out the caveats. Indeed, most databases and datasets including the UKBB that we use here have missing or inaccurate entries. We do discuss it in the text, as well as suggest and employ strategies to mitigate these caveats. We now updated the text to highlight these issues even further. Specifically, in the second paragraph of the “Results” section, we added the following text: “Most large human databases and datasets, including UKBB, have certain limitations, such as incomplete or missing data points. Therefore, before proceeding to modelling aging, we needed to address the following three issues:”

(2) There is no validation dataset to verify the proposed model. The authors suggested that human biological age can be predicted with high accuracy using 12 simple physiological measurements. It will be super useful and convincing if another biobank dataset containing those 12 traits can be applied to the current model.

Thank you for this comment. Indeed, having a replication cohort would be quite valuable. As of today, there is no comparable dataset to verify performance of the clock model or to attempt to validate GWAS results. The closest possible is the NIH-led research program “All Of Us”, which aims to collect data on 1 million people, which unfortunately is not available to for-profit companies. It is theoretically possible to rebuild a clock only using a small number of phenotypes present in both datasets with the goal of training it on one dataset and test-applying it to another, but this won’t ultimately address the accuracy of the wholistic physiological clock presented here. We hope academic labs will utilize our clock-modeling approach and apply it to datasets currently unavailable to us and publish their findings.

To strengthen the credentials of our biological clock, we would like to remind the reviewer that we performed 10 rounds of validation, where, in each round, 10% of the data were left out from the model training such that the clock was created using remaining 90%. The model was subsequently tested on the 10% that was left out. Over 10 rounds, different 10% of data were left out and statistics for this 10-fold cross-validation age available in the supplementary materials. We have now updated the text to make this validation more apparent.

Specifically, we added to the "Results” section, “A mathematical model to predict age” subsection, third paragraph, the following text: “Specifically, we performed 10 rounds of cross-validation, where 10% of data were held out and the remaining 90% used for training. Over 10 rounds, different 10% were held out for validation. In each case, the findings were validated in the test set. Full statistics and approach are described in supplementary computational methods.”

Additionally, the details of this cross-validation are described in detail in supplementary methods.

Additionally, we compared published GWAS results obtained for human aging clocks using modalities that were different yet relevant to human health. Specifically, we looked at GWAS for “Epigenetic Blood Age Acceleration” (Lu et al., 2018), ML-imaging-based human retinal aging clock (Ahadi et al., 2023), PhenoAgeAcceleration and BioAgeAcceleration (Kuo et al., 2021), and the ∆Age GWAS that we presented in our manuscript. We now describe the results of this comparison in our manuscript. Briefly, there is no overlap between GWAS results for any two of these published clocks built via different modalities – retina, DNA methylation, or physiological functions (between each other or with our model). However, there is a significant genetic overlap (p<10E-8) between clocks built using human phenotypic measures in a cohort of National Health and Nutrition Examination Survey (NHANES) III in the United States (7 variables) and ∆Age from Physiological clock from UKBB that we describe here (121 variables), further validating our approach. It is interesting to consider the reasons why genetic associations for human aging built using different modalities do not appear to have common genetic corelates, something we also now discuss in our manuscript.

Specifically, we added to the "Results” section, “Genetic loci associated with biological age” subsection, third paragraph, the following text: “Additionally, we compared our ∆Age GWAS association results with similar GWAS studies that were performed for other biological clocks. For example, (McCartney et al., 2021) used DNA methylation data on 40,000 individuals to compute biological age called GrimAge. After that they calculated an intrinsic epigenetic age acceleration (IEAA, a value similar to ∆Age, which measured a deviation of biological age from chronological age) and performed GWAS.” Additionally, we added to the “Discussion” section, “Broader implications of the model for physiological aging” subsection, fourth paragraph, the following text: “To further analyze the meaning of genetic associations with ∆Age that we described above, we compared several published GWAS results obtained for human aging clocks using different health modalities. Specifically, we looked at GWAS for “Epigenetic Blood Age Acceleration” (Lu et al., 2018), ML-imaging-based human retinal aging clock (Ahadi et al., 2023), PhenoAgeAcceleration and BioAgeAcceleration (Kuo et al., 2021), and the ∆Age GWAS we presented in our manuscript. Surprisingly, we discovered that there is no overlap between GWAS results for any two of these clocks built via different modalities – retina, DNA methylation, or physiological functions. However, there is a significant genetic overlap between clocks built using human phenotypic measures and our ∆Age model we describe. For example, the Biological Age Clock Acceleration calculated using HbA1c, Albumin, Cholesterol, FEV, Urea nitrogen, SBP, and Creatinine (Levine, 2013) in a US cohort [from National Health and Nutrition Examination Survey (NHANES)] yielded 16 significant hits in the GWAS analysis, five of which were also significant in our GWAS for UKBB based ∆Age. These five common loci were close to the following genes - APOB, PIK3CG, TRIB1, SMARCA4, and APOE. The significance of this overlap is p < 10^-8^, suggesting that the ∆Age model we propose might be translatable to other cohorts of people.

An interesting question to consider is why GWAS results from other clock modalities, such as DNA methylation and retinal imaging do not yield any genetic similarities to each other or to physiological and biological clocks. It is possible that these modalities of age assessment depend on completely genetically independent biological processes. For example, in a simplified manner - blood composition might be heavily weighted for DNA methylation, vascular structure for retinal scans, and muscle/bone/kidney health for physiological clocks. Data from model organisms suggest the master regulators of aging exist, and APOE is the best genetic variant known to influence human aging. Interestingly, only the biological and physiological clock models that we propose here pick it up as a hit. Alternatively, it is also possible that the true master regulators of aging rate are under stringent purifying selection; for example, due to an important role in development, and therefore, do not have genetic variability in human populations examined. As such, they could not be identified as hits in any GWAS studies.”

**Reviewer #2 (Public Review):**
In this manuscript, Libert et al. develop a model to predict an individual's age using physiological traits from multiple organ systems. The difference between the predicted biological age and the chronological age -- ∆Age, has an effect equivalent to that of a chronological year on Gompertz mortality risk. By conducting GWAS on ∆Age, the authors identify genetic factors that affect aging and distinguish those associated with age-related diseases. The study also uncovers environmental factors and employs dropout analysis to identify potential biomarkers and drivers for ∆Age. This research not only reveals new factors potentially affecting aging but also shows promise for evaluating therapeutics aimed at prolonging a healthy lifespan. This work represents a significant advancement in data-driven understanding of aging and provides new insights into human aging. Addressing the points raised would enhance its scientific validity and broaden its implications.

Thank you!

Major points:(1) Enhance the description and clarity of model evaluation.The manuscript requires additional details regarding the model's evaluation. The authors have stated "To develop a model that predicts age, we experimented with several algorithms, including simple linear regression, Gradient Boosting Machine (GBM) and Partial Least Squares regression (PLS). The outcomes of these approaches were almost identical". It is currently unclear whether the 'almost identical outcomes' mentioned refer to the similarity in top contribution phenotypes, the accuracy of age prediction, or both. To resolve this ambiguity, it would be beneficial to include specific results and comparisons from each of these models.

Thank you for this comment. We now describe details of the model selection and provide data on outcome caparisons. Briefly, different approaches have different advantages and limitations; however, we chose one approach, and did not develop and analyze several independent models in parallel in order to not artificially inflate our False Discovery Rate (FDR). However, we now provide rationale and comparative performance of these three approaches. Specifically, we added to the "Results” section, “A mathematical model to predict age” subsection, first paragraph the following text: “Different approaches have different advantages and limitations; however, we decided to choose one approach, and not develop and analyze several independent models in parallel in order to not artificially inflate the False Discovery Rate (FDR). We ultimately selected PLS regression because it enabled us to determine the number and composition of components required to predict age optimally from the data, which provides additional insights into the biology of human aging. But before making this selection, we compared the performance of the three approaches. The outcomes of PLS and linear regression were almost identical (R-squared between ∆Age values derived by these two methods was 0.99, meaning that if one model were to predict an individual was 62 years old, the other model would have the same prediction). This similarity is likely due to the small number of predictors (121 phenotypes) and comparatively large number of participants (over 400,000). The correlation between GBM model outcomes and PLS (and linear regression) was slightly smaller (R-squared = 0.87). The reason for the lower correlation is likely the need for imputation in PLS and linear regression models. The GBM model tolerates missing data, whereas linear regression and PLS methods require imputation or removal of individuals with too many datapoints missing, an approach we describe in more detail below.”

Additionally, after we obtained associations of ∆Age values with genetical loci, which formed the candidate base for gene targets to influence human aging (figure 5b), we verified the top association obtained via the PLS model in Linear and GBM models. All the top candidates that we verified had statistically significant associations in all the models of ∆Age (CST3, APOE, HLA locus, CPS1, PIK3CG, IGF1). The precise strengths of the associations were different, but that is to be expected given that linear datasets had some data imputed while GBM model was built with missing values. We believe that due to small number of predictors (121) compared to a vastly larger number of individuals (over 400,000), the differences the three models introduced to final outcomes were quite small.

To convey this message, we added to the "Discussion” section, “Broader implications of the model for physiological aging” subsection, 7th paragraph, the following text: “It is interesting to note that the three approaches we used to generate age prediction model (PLS, GBM, and linear regression) yielded very similar or identical results in performance. We chose to settle on one approach (PLS) to not artificially inflate the False Discovery Rate (FDR); however, we verified that the top genetic loci associations obtained via the PLS model were also obtained in the GBM and linear models. Specifically, the top candidates (CST3, APOE, HLA locus, CPS1, PIK3CG, IGF1) identified in the PLS approach had statistically significant associations in all the models of ∆Age. It is likely that due to the small number of predictors (121) compared to a vastly larger number of individuals (over 400,000), the differences that these models introduce to final outcomes are quite small, which increases our confidence in the results.”

Furthermore, the authors mention "to test for overfitting, a PLS model had been generated on randomly selected 90% of individuals and tested on the remaining 10% with similar results". To comprehensively assess the model's performance, it is crucial to provide detailed results for both the test and validation datasets. This should at least include metrics such as correlation coefficients and mean squared error for both training and test datasets.

Thank you for bringing up this point. The detailed description, details and statistics of cross-validation procedure is described in supplementary computational methods. Briefly, across 10 rounds of validation the Root Mean Square Error of Prediction (RMSEP) did not exceed 4.81 for females when all 9 PLS components were considered, and RMSEP form males was 5.1 when all 11 components were considered. The variation of RMSEP between different datasets was less than 0.1. We have now updated the text to make this validation more apparent. Specifically, we added to the "Results” section, “A mathematical model to predict age” subsection, third paragraph the following text: “Specifically, we performed 10 rounds of cross-validation, where 10% of data were held out and the remaining 90% used for training. Over 10 rounds, different 10% were held out for validation. In each case, the findings were validated in the test set. Full statistics and approach are described in supplementary computational methods.”

(2) External validation and generalization of resultsTo enhance the robustness and generalizability of the study's findings, it is crucial to perform external validation using an independent population. Specifically, conducting validation with the participants of the 'All of Us' research program offers a unique opportunity. This diverse and extensive cohort, distinct from the initial study group, will serve as an independent validation set, providing insights into the applicability of the study's conclusions across varied demographics.

Thank you for this comment. As we mentioned above, we agree that having a replication cohort would be very valuable for this study, as well as many other studies that stem from UKBB dataset. However, yet, there is no comparable dataset to verify performance of the clock or to attempt to validate GWAS results. The closest possible is NIH-led research program “All Of Us”, which aims to collect data on 1 million people, which unfortunately is not available to for-profit companies. It is theoretically possible to rebuild a clock only using the small number of phenotypes present in both datasets with the goal of training it on one dataset and test-applying it to another, but that approach would not ultimately be informative about the accuracy of the complete physiological clock presented here. We hope academic labs will utilize our clock approach and apply it to datasets currently unavailable to us and publish their findings. For the detailed response on this issue, please see the response to the second comment of the first reviewer above.

**Recommendations for the authors:**

**Reviewer #1 (Recommendations For The Authors):**
Specific questions/suggestions:- It looks like the ages of participants are enriched around 60 years (Fig. 1, Fig 3b). Can authors clarify whether age distribution affects the correlation tests (e.g. correlation in Fig 2)?

Indeed, the distribution of people by age is enriched by 60–65-year-olds and is depleted at younger and older ages. Such a distribution influences the uncertainty of correlations that we compute, with error bars being larger for 40- and 70-year-olds and lower for 50- and 60-year-olds. The example of this can be seen on figure 1F. Figures 2a,b,g,h mostly deal with the correlation of phenotypes with each other and thus are not influenced by age. For other computations, such age prediction, it is theoretically possible that if age determinants among 65-year-olds differ from those for 40- or 80-year-olds, the calculated contributions would be skewed to increase accuracy in the middle of distribution at the expense of the ends. ∆Age, however, was explicitly normalized for each age cohort (Fig. 3a) to avoid “birth cohort” bias, therefore minimizing the effect of uneven distribution on further analysis, such as GWAS. We now acknowledge and describe this feature of UKBB dataset in the first paragraph of the “Results” section.

- Phenotypic variation usually increases during aging. However, the authors showed that delta-age and age are not correlated (Figure 3a), suggesting that biological variation does not increase during aging in their analysis. Can authors provide more evidence supporting their findings? Is this phenomenon affected by their normalization method?

Thank you for this comment. We find that there is no strict rule for phenotypic variation change with age. Certain phenotypes, such as blood pressure (Fig. 1a) or SHGB (Fig. 1d), indeed increase in variation with advanced age, however many others, such as grip strength (Fig. 1b) and BMI do not change in variation, and certain phenotypes even decrease their variation with age. As we stated above, in order to minimize the possible effect of “birth cohort” bias on subsequent analysis, as well as uneven distribution of people across ages, ∆Age was normalized per age cohort. Additionally, purifying selection likely also limits how far most physiological factors can deviate. For example, people with too high or too low blood pressures would simply perish, which would limit continuous increase in variation.

- Authors correlate GWAS data with delta-age (Figure 4). It would be important to show whether the delta-age from young and old participants correlates with GWAS patterns in a similar manner. If not, the authors have to consider how age differences affect delta-age and the GWAS correlation. For example, the authors mentioned that APOE genotype influences age-delta even in the 40-year-old group (Figure 4f). If the APOE genotype already shows high delta-age in the 40-year-old group, how does aging affect the delta-age distribution?

Thank you for this comment. It is an interesting question to understand how age influences GWAS hits identified through ∆Age. At the same time, one must remember that our dataset is cross-sectional in nature and “different age” in reality is a subset of different people, which lived in different times with different exposures to environments and different standards of medical care (which are evolving over time). We specifically attempted to factor age and this “cohort effect” out of our analysis and presented Figure 4f simply as an illustration that APOE variants seem to influence human aging at any age, which challenges the theory proposed by previous studies that APOE is implicated in aging simply because APOE4 carriers likely die from Alzheimer disease and are thus excluded from the oldest cohorts. To investigate the question raised by the reviewer it is possible to do GWAS on age, however one must keep in mind the limitations associated with interpreting those results; as “age” in reality (in this cross-sectional cohort) also represents changes in population composition, changes in the environment, food quality, early life care, medical care, social habits, and other parameters associated with changing society.

- For the discussion part, it would be great if the authors could add one section to provide guidelines for future human and lab animal studies based on observations from the current study. For example, what physiological traits are most useful, and what can be further added when collecting human data?

Thank you for the great suggestion. We now propose and discuss certain experiments that can be performed in humans and animals to better differentiate between drivers and markers of aging.

- In line 479, I found the statement "It is possible that synapse function accounts for the association of computer gaming with ΔAge" came from nowhere, and suggest removing it.

Done—thank you.

- Minor. Line 155. Is it a wrong citation of table S2c, 2d as there are only 2a and 2b?

Thank you, corrected.

**Reviewer #2 (Recommendations For The Authors):**
(1) Between lines 300-305, there is a missing reference to Figure 3e.

Thank you, corrected.

(2) For Figures 4a and 4c, please add the lambda statistic to the QQ plots.

Thank you, we have added lambda inflation factors to the QQ plots.

(3) In line 384, the p-value cut-off is mentioned as 10-9. However, this does not seem to be consistently represented in Figures 4b and 4d, where the gray lines do not align with this threshold. Please adjust these figures to accurately reflect the mentioned p-value cut-off.

Thank you, corrected.

(4) Clarification for Figure 5a. Add titles and correlation coefficients to Figure 5a to clearly define what the clusters represent. Please also add a discussion to explain why the cluster 10 (general health) dropout model can affect ∆Age compared to the full model, with some individuals showing a 5-year difference. Furthermore, despite the substantial effect of removing cluster 10 on ΔAge, all the top loci remain unchanged in terms of effect sizes and p-values compared to the full model.

We have added the titles and correlation coefficients to the Figure 5a. Thank you for these suggestions, it makes the presentation of data much clearer. It is an interesting observation that whereas dropping out cluster 10 resulted in quite significant changes of ∆Age distribution, the genetic signature as determined by GWAS did not change much. The most obvious explanation is that many parameters in this category are influenced by environment more than by genetics, therefore genetic signature did not change much after the cluster removal. We now mention this observation in the text. Specifically, in the subsection “Cluster-dropout analysis enriches for GWAS hits that influence aging globally”, we added the following text: “Another interesting observation is that degree by which certain cluster contributes to the model does not necessarily correlate with how much this cluster contributes to genetic signature of human aging. For example, while dropping out cluster 10 (General Health) resulted in quite significant changes of ∆Age distribution (R^2^=0.88), the genetic signature as determined by GWAS did not change substantially. The most likely explanation is that many parameters in this category are influenced by environment more strongly than by genetics; for example, not as much as caused by cluster 1 (muscle-related) removal.”

(5) Discussion on drivers and markers. Given the theoretical nature of the study, it would be beneficial to propose potential experimental validations for your findings. Even if these validations have not been performed, suggesting them would greatly enhance the value of the discussion.

Thank you, it is a great idea. We now propose and discuss certain experiments that can be performed in humans and animals to better differentiate between drivers and markers of aging. Specifically, in the subsection “Cluster-dropout analysis enriches for GWAS hits that influence aging globally”, we added the following text: “To definitively distinguish whether a gene is a driver or a marker of aging, an experiment would need to be performed. It is possible that certain gene activities are influenced by existing FDA-approved medications, and retrospective analyses of human cohorts who take certain medications can be performed. More likely, however, an animal model would need to be employed, where animals with candidate genes modified via genetic means are investigated for lifespan and onset and progression of age-associated conditions. For example, one can engineer a mouse with a conditional allele of Cystatin-C and evaluate how changes in dosage of this protein influence various phenotypes of aging.”